# BECLIN1 is essential for intestinal homeostasis involving autophagy-independent mechanisms through its function in endocytic trafficking

Sharon Tran[1,2,13], Juliani Juliani[1,2,3,4,13], Tiffany J. Harris[1,2], Marco Evangelista[1,2], Julian Ratcliffe [5], Sarah L. Ellis [1,2], David Baloyan[1,2], Camilla M. Reehorst [1,2], Rebecca Nightingale[1,2], Ian Y. Luk[1,2], Laura J. Jenkins[1,2], Sonia Ghilas[1,2], Marina H. Yakou [1,2], Chantelle Inguanti[1,2], Chad Johnson[5], Michael Buchert [1,2], James C. Lee[6,7], Peter De Cruz[8,9], Kinga Duszyc[10], Paul A. Gleeson[11], Benjamin T. Kile[12], Lisa A. Mielke [1,2], Alpha S. Yap [10], John M. Mariadason [1,2], W. Douglas Fairlie [1,2,3,4] ✉ & Erinna F. Lee [1,2,3,4] ✉

Autophagy-related genes have been closely associated with intestinal homeostasis. BECLIN1 is a component of Class III phosphatidylinositol 3-kinase complexes that orchestrate autophagy initiation and endocytic trafficking. Here we show intestinal epithelium-specific BECLIN1 deletion in adult mice leads to rapid fatal enteritis with compromised gut barrier integrity, highlighting its intrinsic critical role in gut maintenance. BECLIN1-deficient intestinal epithelial cells exhibit extensive apoptosis, impaired autophagy, and stressed endoplasmic reticulum and mitochondria. Remaining absorptive enterocytes and secretory cells display morphological abnormalities. Deletion of the autophagy regulator, ATG7, fails to elicit similar effects, suggesting additional novel autophagy-independent functions of BECLIN1 distinct from ATG7. Indeed, organoids derived from BECLIN1 KO mice show E-CADHERIN mislocalisation associated with abnormalities in the endocytic trafficking pathway. This provides a mechanism linking endocytic trafficking mediated by BECLIN1 and loss of intestinal barrier integrity. Our findings establish an indispensable role of BECLIN1 in maintaining mammalian intestinal homeostasis and uncover its involvement in endocytic trafficking in this process. Hence, this study has important implications for our understanding of intestinal pathophysiology.

Autophagy is an evolutionarily conserved process critical for the maintenance of eukaryotic cellular homeostasis. It encapsulates cytoplasmic material within autophagosomes and mediates their delivery to the lysosome for degradation. Critically, genome-wide association studies (GWAS) have identified that polymorphisms in genes that regulate autophagy correlate with a susceptibility to inflammatory bowel disease (IBD), which includes Crohn's disease and ulcerative colitis[1]. Subsequent studies aiming to understand this connection between autophagy and intestinal health led

[1]Olivia Newton-John Cancer Research Institute, Heidelberg, VIC, Australia. [2]School of Cancer Medicine, La Trobe University, Bundoora, VIC, Australia. [3]Department of Biochemistry and Chemistry, School of Agriculture, Biomedicine and Environment, La Trobe University, Bundoora, VIC, Australia. [4]La Trobe Institute for Molecular Science, La Trobe University, Bundoora, VIC, Australia. [5]Bioimaging Platform, La Trobe University, Bundoora, VIC, Australia. [6]Genetic Mechanisms of Disease Laboratory, the Francis Crick Institute, London, United Kingdom. [7]Institute for Liver and Digestive Health, Division of Medicine, Royal Free Hospital, University College London, London, United Kingdom. [8]Department of Gastroenterology, Austin Health, Melbourne, VIC, Australia. [9]Department of Medicine, Austin Academic Centre, The University of Melbourne, Melbourne, VIC, Australia. [10]Institute for Molecular Bioscience, The University of Queensland, St. Lucia, Brisbane, QLD, Australia. [11]Department of Biochemistry and Pharmacology and Bio21 Molecular Science and Biotechnology Institute, The University of Melbourne, Melbourne, VIC, Australia. [12]Garvan Institute of Medical Research, Darlinghurst, NSW, Australia. [13]These authors contributed equally: Sharon Tran, Juliani Juliani. ✉e-mail: Doug.Fairlie@onjcri.org.au; Erinna.Lee@latrobe.edu.au

to the generation of mice deficient for, or bearing, disease variants in autophagy regulators. These studies revealed autophagy-related roles intrinsic to intestinal epithelial cells (IECs), including the protection against enteric pathogen infections[2,3], the homeostatic and secretory capacity of specialized IECs, namely Paneth and goblet cells[4-10], and the survival of intestinal stem cells (ISCs)[11]. Beyond IECs, autophagy also contributes to the survival and function of intestinal stromal cells, including immune cells, where disruptions to these autophagic functions have adverse impacts on the intestinal cytokine and growth factor milieu[12,13]. Whilst these are important roles for autophagy regulators in the maintenance of intestinal homeostasis, autophagy-deficient intestines generally function normally, and autophagy deficiency alone is insufficient for driving spontaneous intestinal pathologies, consistent with the paradigm that IBD pathogenesis is multifactorial[14].

BECLIN1 is a regulator of autophagy that exerts its cellular functions as a scaffolding subunit of two mutually exclusive Class III phosphatidylinositol 3-kinase (PI3KC3) complexes[15]. These complexes share two common subunits in addition to BECLIN1: the lipid kinase, Class III PI3KC3, and the regulatory subunit, VPS15. Distinguishing the two complexes are autophagy-related protein 14 (ATG14) in Complex 1 (PI3KC3-C1) and UV radiation resistance gene product (UVRAG) in Complex 2 (PI3KC3-C2)[16-19]. Both complexes promote autophagosome formation and maturation, although notably, PI3KC3-C2 is additionally localized to endosomes and, therefore, can also regulate endocytic trafficking[17,18]. There is some indication that BECLIN1 may be relevant to intestinal health. For example, using a cell culture model, BECLIN1 was shown to play a constitutive autophagy-independent role in intestinal tight junction barrier function through the endosomal degradation of occludin[20]. In *Drosophila* ISCs, the orthologue of BECLIN1, ATG6, is required for maintaining the integrity of this cellular compartment in response to aging[21]. Recently, it was found that mice expressing constitutively active mutant BECLIN1 have a thicker colonic mucosal layer due to autophagy-mediated alleviation of endoplasmic reticulum (ER) stress, demonstrating a potential role for BECLIN1 in the maintenance of intestinal homeostasis[22]. However, despite these indicators, and the plethora of genetic models in which other key autophagy regulators have been deleted in the intestinal epithelium[14], to date, there is no published murine model in which BECLIN1 has been genetically deleted in this compartment.

In this study, we report for the first time that severe fatal intestinal disruption ensues following intestinal epithelial-specific deletion of BECLIN1 in adult mice. This phenotype is not seen when the autophagy regulator ATG7 is deleted. We postulate that the intestinal architectural disruption, extensive IEC death, and functional abnormalities leading to the fatal crippling of intestinal homeostasis following BECLIN1 loss also involve autophagy-independent mechanisms caused by defective endocytic trafficking.

## Results

### Generation of conditional *Becn1* and *Atg7* intestinal epithelium-specific inducible knock-out mice

To induce intestinal epithelium-specific deletion of *Becn1*, *Becn1*^{fl/fl} mice were bred to mice expressing tamoxifen-inducible Cre recombinase from the villin 1 promoter (*Vil1-CreERT2*, Fig. 1a). For comparison, we also generated mice in which the pro-autophagic E1-like ligase ATG7, which unlike BECLIN1 has no reported roles in endocytic trafficking[23], could be deleted in a similar manner (Fig. 1a). The resulting tamoxifen-treated *Becn1*^{fl/fl};*Vil1-CreERT2*^{Cre/+} and *Atg7*^{fl/fl};*Vil1-CreERT2*^{Cre/+} mice will hereafter be referred to as *Becn1*^{ΔIEC} and *Atg7*^{ΔIEC} mice respectively, whilst their littermate tamoxifen-treated *Becn1*^{+/+};*Vil1-CreERT2*^{Cre/+} and *Atg7*^{+/+};*Vil1-CreERT2*^{Cre/+} mice are referred to as *Becn1*^{wtIEC} and *Atg7*^{wtIEC} mice respectively (Fig. 1a). All time points, unless otherwise specified, indicate the time after receipt of the first tamoxifen dose. Inducible conditional deletion of either *Becn1* or *Atg7* in the intestinal epithelium was confirmed by genotyping (Supplementary Fig. 1) and Western blotting (Fig. 1b) of IECs isolated from *Becn1*^{ΔIEC} and *Atg7*^{ΔIEC} mice. Both showed significantly

reduced BECLIN1 and ATG7 expression, respectively, compared to control wild-type mice after seven days. As expected, loss of either BECLIN1 or ATG7 resulted in disrupted basal autophagic flux as indicated by an accumulation of p62 and LC3 protein levels, or an increased LC3-I:LC3-II ratio, in *Becn1*^{ΔIEC}- and *Atg7*^{ΔIEC}-derived IECs (Fig. 1b). Our successful generation of mice in which either BECLIN1 or ATG7 can be specifically deleted in the intestinal epithelium in an inducible manner, enabled us to investigate the role of BECLIN1 in this tissue. In addition, the comparison with ATG7 deleted mice enabled the delineation of the contributions of autophagy *versus* endocytic trafficking mechanisms to this process.

### Deletion of BECLIN1, but not ATG7, in the intestinal epithelium, leads to severe fatal loss of intestinal homeostasis

Strikingly, within seven days post-induction, *Becn1*^{ΔIEC} mice developed severe and fatal enteritis with 100% penetrance with the small intestine appearing lytic, necrotic, and grossly swollen (Fig. 1c, d). These mice presented with shortened small intestinal, but not colon, lengths (Fig. 1d, e) and significant body weight loss (>10%, Fig. 1f), with humane euthanasia being required. Inspection of all other organs, including the colon, at this time point, revealed they were grossly anatomically normal and similar to their wild-type littermates. Notably, this phenotype contrasted with *Atg7*^{ΔIEC} mice over the same time course (Fig. 1c–f); these mice did not develop any fatal or macroscopic intestinal pathologies when the same parameters were assessed, even when aged up to one month (Supplementary Fig. 2). However, interestingly, at this time point, they displayed a small but significant decreased weight gain and increased intestinal length compared to wild-type controls (Supplementary Fig. 2).

Characterization of the intestinal pathology of *Becn1*^{ΔIEC} mice by haematoxylin and eosin (H&E) staining revealed dramatic architectural changes, which included the loss of intestinal morphological integrity with villi blunting (scored using Erben et al.[24] classifications (Fig. 1g, h), and evidence of mucosal and submucosal, though no transmural, immune infiltration (Fig. 1g, i). Staining of intestinal sections with cleaved caspase-3 (CC3) and terminal deoxynucleotidyl transferase dUTP nick-end labeling (TUNEL) showed widespread apoptosis of epithelial cells both in the crypt and along the villi length (Fig. 1j). Flow cytometry-based immunophenotyping of intraepithelial lymphocytes (IELs) isolated from these mice showed an increased proportion of total lymphocytes, specifically T cells (Supplementary Fig. 3). In particular, there was a statistically significant increase in the numbers of cytotoxic CD8 + T cells. There was also a trend towards increased gamma-delta T cells important for immune surveillance at epithelial and mucosal surfaces, though this was not statistically significant. Consistent with the dramatic morphological disruption and loss of IEC survival in *Becn1*^{ΔIEC} mice, we observed a severely compromised total intestinal barrier as determined by the increased release of fluorescein isothiocyanate-dextran (FITC-dextran) from the intestinal lumen into the circulation of these mice (Fig. 1k). Importantly, these dramatic intestinal disruptions were not observed in *Atg7*^{ΔIEC} mice over the same time frame (Fig. 1g–k, Supplementary Fig. 3).

Taken together, our results demonstrate an essential role for BECLIN1 in the maintenance of intestinal homeostasis. We postulate that its autophagy-independent function in endocytic trafficking likely contributes substantially given the striking intestinal phenotypic differences between *Becn1*^{ΔIEC} and *Atg7*^{ΔIEC} mice.

### BECLIN1 is essential to the survival and function of specialized intestinal epithelial cells

The intestinal epithelium is made up of a single layer of cells comprising differentiated IEC subpopulations, all of which arise from ISC progenitors in the crypt base. Each IEC subpopulation confers their own distinct specialized function, which together regulate intestinal homeostasis[25]. Staining for the proliferation marker Ki-67 in *Becn1*^{ΔIEC} crypts and transit-amplifying regions was retained on day 7, as in *Atg7*^{ΔIEC} and control mice (Fig. 2a),

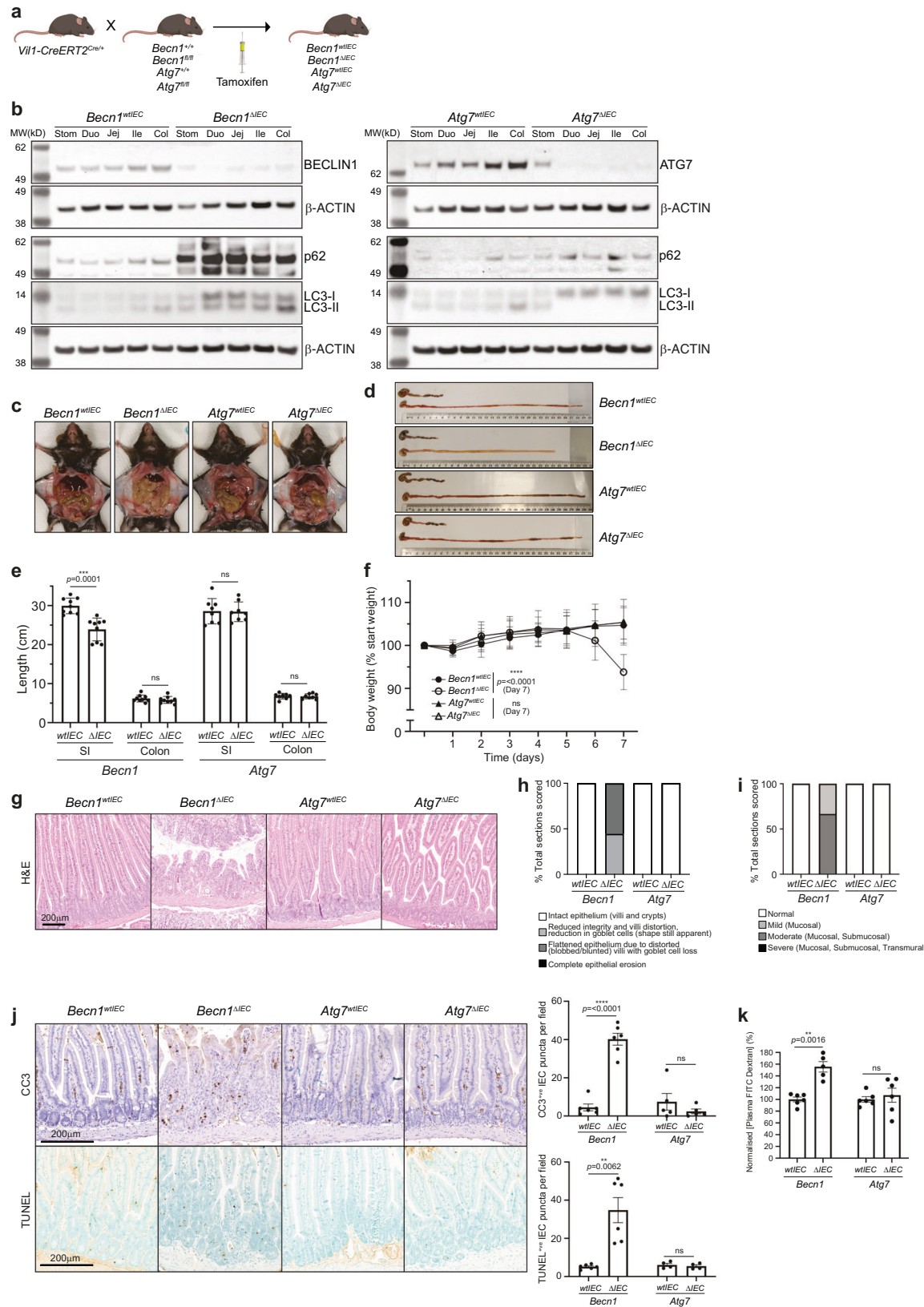

indicating that the proliferative capacities of ISCs and transit-amplifying progenitor cells (TACs) are not adversely impacted by BECLIN or ATG7 loss at this time point.

To interrogate any IEC-specific changes within the intestinal epithelium following BECLIN1 or ATG7 deletion on day 7, we assessed each of the

major IEC subtypes in our knock-out mice using immunohistological techniques. We observed a significant loss of specific IEC subpopulations, in $Becn1^{\Delta IEC}$ but not $Atg7^{\Delta IEC}$ or wild-type littermates. Specifically, there were significant reductions in goblet (PAS/AB[+ve], Fig. 2b) and tuft (DCLK1[+ve], Fig. 2c) cell numbers. In contrast, there were no significant changes to the

**Fig. 1 | The absence of BECLIN1, but not ATG7, leads to fatal loss of intestinal homeostasis. a** Intestinal epithelium-specific deletion of loxP-flanked *Becn1* or *Atg7* in adult mice is mediated by tamoxifen-inducible Cre recombinase from the villin 1 promoter. **b** Deletion of either BECLIN1 or ATG7 leads to defective autophagy as indicated by increased levels of total p62 and LC3, or increased LC3-I relative to LC3-II. Representative images of **c** abdominal necropsy and **d** intestinal tracts demonstrating the severe intestinal disruption seen only in the absence of BECLIN1 (upper: colon; lower: small intestine). **e** Intestinal lengths of mice at endpoint. **f** Body weight of mice over time normalized to day 0. **g** FFPE sections of mouse intestines, stained with H&E, reveal the extensive loss of the characteristic villus-crypt structures of the small intestines from *Becn1^ΔIEC^* mice. **h** Scoring of intestinal morphological integrity. **i** Loss of intestinal homeostasis in the absence of BECLIN but not ATG7 leads to increased mucosal and/or submucosal immune cell infiltration. **j** There was evidence of increased apoptosis, as determined by cleaved caspase-3 (CC3) and terminal doxynucleotidyl transferase dUTP nick-end labeling (TUNEL) staining, along the crypt-villus axis in *Becn1^ΔIEC^* mice only. **k** Loss of BECLIN1 results in loss of intestinal barrier integrity, leading to increased permeability to FITC-Dextran. For all data represented, except for the histology images, at least *n* = 6 biologically independent mice of each genotype were used. For the histology images, at least *n* = 4 biologically independent mice per genotype was used. Data represent *n* = 3 independent experiments unless otherwise indicated. Graphs show the mean ± S.E.M, except for the small intestinal lengths and body weight graphs, which show the mean ± S.D. Significance was determined by Welch's unpaired *t* test. ns not significant ($p > 0.05$). Stom stomach, Duo duodenum, Jej jejunum, Ile ileum, Col colon, SI small intestine.

numbers of enteroendocrine (ChgA^+ve, Fig. 2d) or Paneth (lysozyme^+ve, Fig. 2e) subpopulations.

Of the IEC subpopulations that remained significantly unchanged in numbers, there were instead observable abnormalities in their morphologies suggestive of functional disruptions pertaining to their anti-microbial capacities. Specifically, inspection of *Becn1^ΔIEC^* Paneth cells revealed drastic morphological changes, with an increased proportion of cells being "depleted" (using the lysozyme staining pattern allocations published by Cadwell et al.[6]), and containing fewer granules and reduced lysozyme levels compared to wild-type mice (Fig. 3a, b). Other Paneth cells appeared swollen with aberrantly accumulated lysozyme-positive granules, which we have newly assigned as "swollen granular" (Fig. 3a, b). Some swollen cells also displayed a "ruptured" appearance, which we have further categorized as "swollen diffuse" (Fig. 3a, b). Whilst Paneth cells from *Atg7^ΔIEC^* mice also demonstrated altered granule integrity with a shift to a more disordered patterning which is consistent with published reports (Fig. 3a, b)[8,26], these defects were not as extensive as those observed in *Becn1^ΔIEC^* mice.

In addition to Paneth cells, we also observed defects in staining for intestinal alkaline phosphatase (IAP), a component of the gut mucosal defense system normally expressed at the brush border of villus-associated enterocytes. Levels of IAP were visibly reduced, particularly towards the distal end of the small intestine, in *Becn1^ΔIEC^* but not *Atg7^ΔIEC^* or wild-type mice (Fig. 3c). Furthermore, even though staining in the duodenum at the proximal end of the small intestine was still evident, there was noticeable mislocalization of IAP, shifting from a well-defined border staining to a more diffuse cytoplasmic distribution (Fig. 3d). Ultrastructural examination of *Becn1^ΔIEC^* enterocytes by TEM found electronlucent tubulated single membrane structures resembling a swollen ER compartment, suggestive of ER stress (Fig. 3e). There was also evidence of degenerating mitochondria (Fig. 3e) and complete disappearance of autophagosomes as expected.

Taken together, these results indicate, for the first time, essential roles for BECLIN1 in maintaining the survival and anti-microbial functions of the different specialized epithelial cells that constitute the barrier function of the intestinal epithelium. Importantly, its coordinated regulation of intestinal homeostasis goes far beyond that implicated for the autophagy regulator ATG7[23].

## BECLIN1 deletion in small intestinal organoids leads to increased apoptosis and morphological defects

Our results indicate essential IEC-intrinsic contributions made by BECLIN1 to the heterogenous, differentiated cell-types of the small intestine and for the maintenance of intestinal integrity and homeostasis. In order to further probe the mechanisms by which BECLIN1 contributes to these, and given the IEC-intrinsic nature of these functions, we isolated and expanded intestinal crypts from *Becn1^fl/fl^*; *Becn1^+/+^*; *Atg7^fl/fl^*; and *Atg7^+/+^;Vil1-CreERT2^Cre/+^* mice to generate intestinal epithelial organoid cultures. Treatment with the tamoxifen metabolite, 4-hydroxytamoxifen (4-HT) induced deletion of BECLIN1 or ATG7 in *Becn1^fl/fl^;Vil1-CreERT2^Cre/+^* (*Becn1^ΔIEC^*) and *Atg7^fl/fl^;Vil1-CreERT2^Cre/+^* (*Atg7^ΔIEC^*) respectively but not in *Becn1^+/+^;Vil1-CreERT2^Cre/+^* (*Becn1^wtIEC^*) or *Atg7^+/+^;Vil1-CreERT2^Cre/+^* (*Atg7^wtIEC^*) organoids (Supplementary Fig. 4).

Similar to the effects seen in mouse intestine, eight days following the first dose of 4-HT treatment, *Becn1^ΔIEC^* organoids were significantly smaller and displayed reduced numbers of budding crypt-like domains per organoid compared to its vehicle-treated wild-type equivalent (Fig. 4a–c). In contrast, *Atg7^ΔIEC^* organoids did not display the morphological defects seen in the absence of BECLIN1 and were similar in appearance to *Becn1^wtIEC^* and *Atg7^wtIEC^* organoids (Fig. 4a–c). Consistent with the extensive IEC apoptosis, and time to death, seen in the intestinal epithelia of *Becn1^ΔIEC^* mice, we also observed significantly exacerbated apoptosis from day 6 in *Becn1^ΔIEC^* organoids as determined by Annexin V/Propidium Iodide (PI) staining by flow cytometry analysis (Fig. 4d). In contrast, we did not see the same extent of apoptotic death in *Atg7^ΔIEC^*, *Becn1^wtIEC^* and *Atg7^wtIEC^* organoids (Fig. 4d). Notably, the increased apoptosis observed over time in intestinal organoids following BECLIN1 deletion is consistent with the timing of IEC apoptotic death and manifestation of the fatal intestinal phenotype of *Becn1^ΔIEC^* mice.

## BECLIN1 loss leads to mislocalization of E-CADHERIN due to defects in the endocytic trafficking pathway

Thus far, our data show that BECLIN1 loss leads to a compromised intestinal barrier. To provide a functional link between the increased intestinal permeability seen in *Becn1^ΔIEC^* mice (Fig. 1k), and the role of BECLIN1 in endocytic trafficking, we focused on E-CADHERIN, a major constituent of the adherens junction which mediates cell-cell adhesion. Correct cellular localization of E-CADHERIN involves several steps of endocytic trafficking, including RAB5^+ve early endosomes[27] and RAB11^+ve recycling endosomes[28,29]. These trafficking vesicles themselves have been shown to be regulated by BECLIN1[30,31]. Stable E-CADHERIN membrane localization is critical for epithelial barrier integrity, and its loss leads to fatal intestinal disruption in mice[32], strikingly reminiscent of our *Becn1^ΔIEC^* phenotype. Recently, BECLIN1 was also shown to mediate E-CADHERIN surface localization to adherens junctions in breast cancer cells[33]. Due to these multiple connections, we performed whole-mount immunofluorescent staining of E-CADHERIN together with marker proteins of each type of endosome along the endocytic trafficking pathway including RAB5 and EEA1 (early endosomes), RAB7, (late endosomes), and RAB11 (recycling endosomes) in intestinal organoids to gain insights into how BECLIN1 deletion impacts intestinal homeostasis.

Whilst *Becn1^wtIEC^* organoids displayed the expected continuous linear pattern along the lateral cell borders, and predominant localization at the apical surface of E-CADHERIN, its localization in *Becn1^ΔIEC^* organoids was significantly different (Fig. 5a, f, k, Supplementary Figs. 5–9). Here, the prominent and well-defined linear staining along the lateral borders, as well as basal and apical surfaces of the intestinal epithelial cells was lost (Fig. 5a, f, k, Supplementary Figs. 5–9). Instead, punctate cytoplasmic E-CADHERIN staining at the apical end of these cells was observed (Fig. 5a, f, k, Supplementary Figs. 5–9).

In *Becn1^ΔIEC^* organoids, the number of RAB5^+ve early endosomes (Fig. 5a, b, Supplementary Fig. 5) was significantly reduced. They were

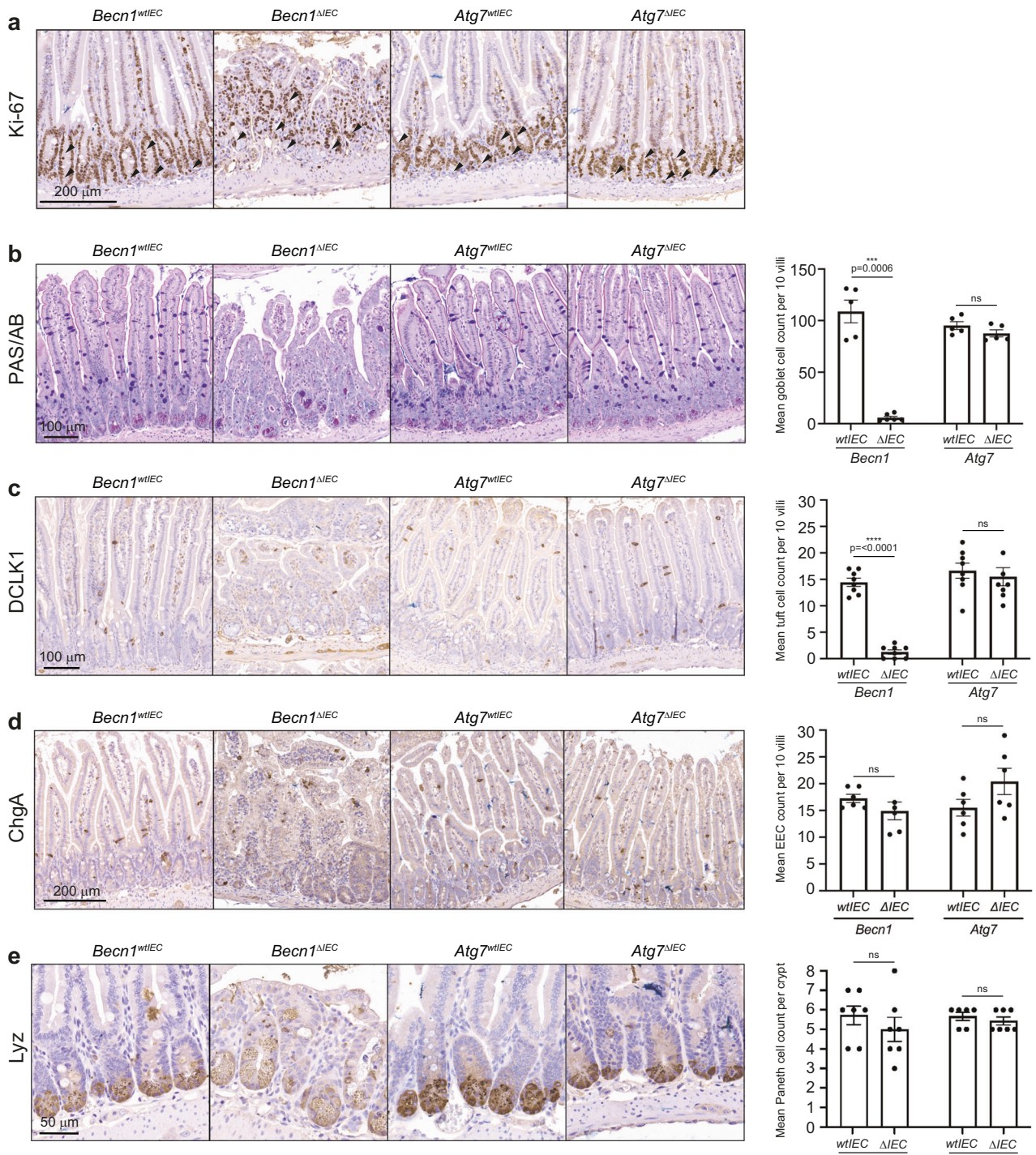

**Fig. 2 | BECLIN1 deletion leads to loss of differentiated intestinal epithelial cell subpopulations. a** The proliferative capacity of cells in the intestinal stem cell compartment and transit-amplifying regions (black arrows), remains intact when BECLIN1 or ATG7 is absent. FFPE sections of small intestines were stained with specific markers for the various IEC subpopulations. BECLIN1 is critical for the maintenance of specific IEC subpopulations such as **b** goblet (Periodic acid-Schiff-Alcian Blue, PAS/AB staining) and **c** tuft (doublecortin like kinase 1, DCLK1, staining) cells, but not others such as **d** enteroendocrine (chromogranin A, ChgA staining) and **e** Paneth (lysozyme, Lyz staining) cells. Quantitation of these specific IECs is represented in graphs on the right. All data represent at least $n = 4$ biologically independent mice per genotype from $n = 3$ independent experiments. Graph shows the mean ± S.E.M. Significance determined by Welch's unpaired t-test. ns = not significant ($p > 0.05$). EEC: enteroendocrine cells.

abnormally enlarged (Fig. 5c) and lost their normal apical-basal uniform distribution seen in wild-type organoids (Fig. 5a, Supplementary Fig. 5) suggestive of disturbances in early endosome maturation. Notably, the cytoplasmic distribution of E-CADHERIN towards the apical end of $Becn1^{\Delta IEC}$ IECs colocalized and appeared trapped within these aberrantly

large and mislocalized RAB5$^{+ve}$ early endosomes (Fig. 5a, d, e, Supplementary Fig. 5). We also examined the early endosomal RAB5 effector protein, EEA1, and found that like RAB5, the defined apical membrane staining of EEA1 staining was lost and instead diffusely localized throughout the cytoplasm towards the apical end of $Becn1^{\Delta IEC}$ IECs (Supplementary

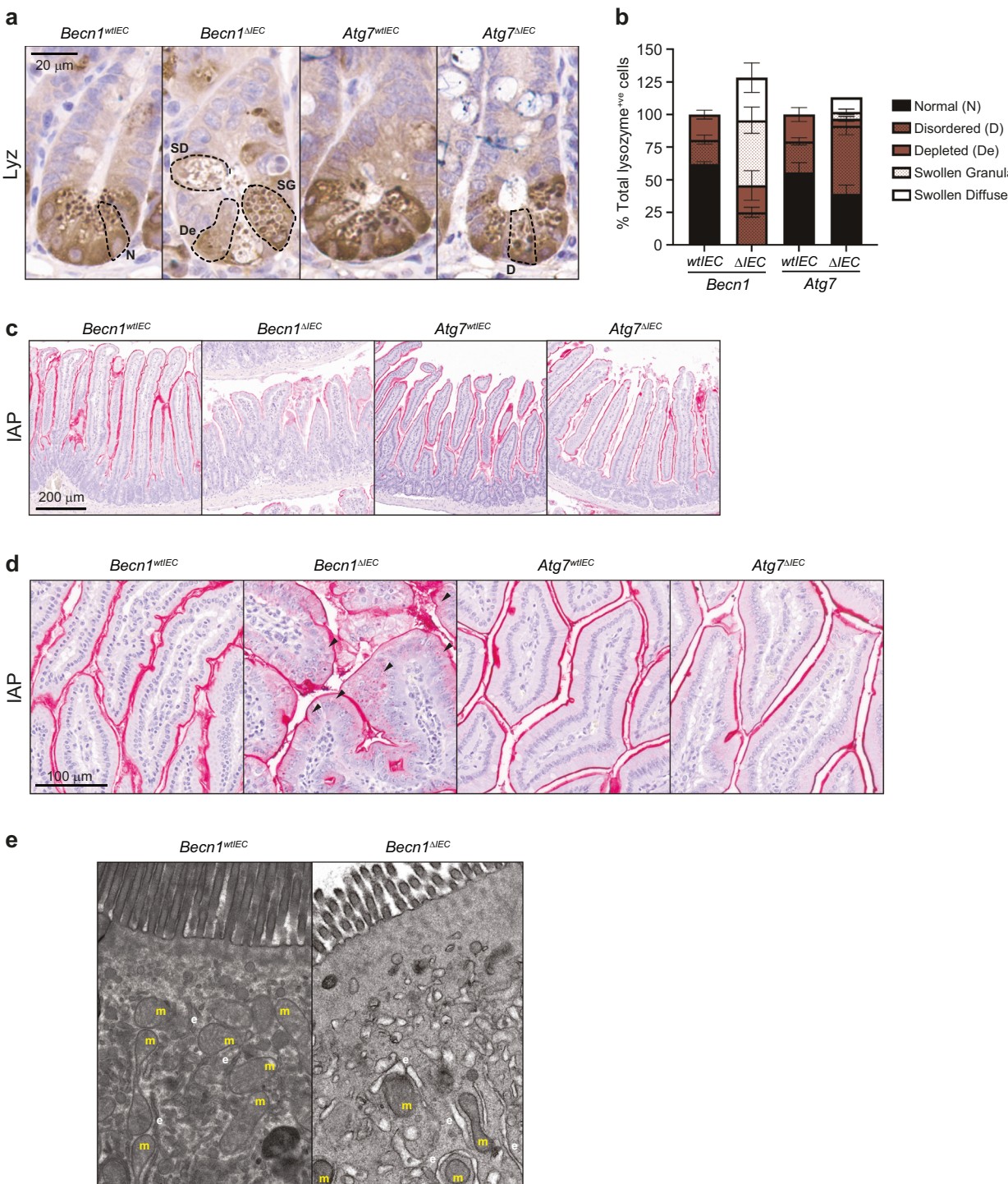

**Fig. 3 | BECLIN1 is critical for maintaining the specialized functions of secretory intestinal epithelial cells.** FFPE sections of small intestines were stained for the antimicrobial proteins lysozyme (Lyz) and intestinal alkaline phosphatase (IAP) produced by both Paneth and enterocytes, respectively. **a** Dashed lines delineating the borders of Paneth cells demonstrate the significant morphological abnormalities of lysozyme-containing secretory granules from both $Becn1^{\Delta IEC}$ and $Atg7^{\Delta IEC}$ animals, though the defects in the absence of BECLIN1 are far more pronounced. **b** Graphical representation of the morphological classifications of abnormal secretory granules in Paneth cells. Villus-associated enterocytes deficient for BECLIN1 demonstrate **c** almost absent IAP expression or where IAP is present, **d** diffuse cytoplasmic staining (black arrows) as opposed to well-defined membrane localization. **e** Representative transmission electron microscopy (TEM) images of $Becn1^{\Delta IEC}$ and $Becn1^{wtIEC}$ enterocytes reveal distorted mitochondria (m) and electron-lucent tubulated structures characteristic of swollen ER (e) and ER stress in the former. All data represent at least $n = 4$ biologically independent mice per genotype from $n = 3$ independent experiments. Graph in **b** shows the mean ± S.D.

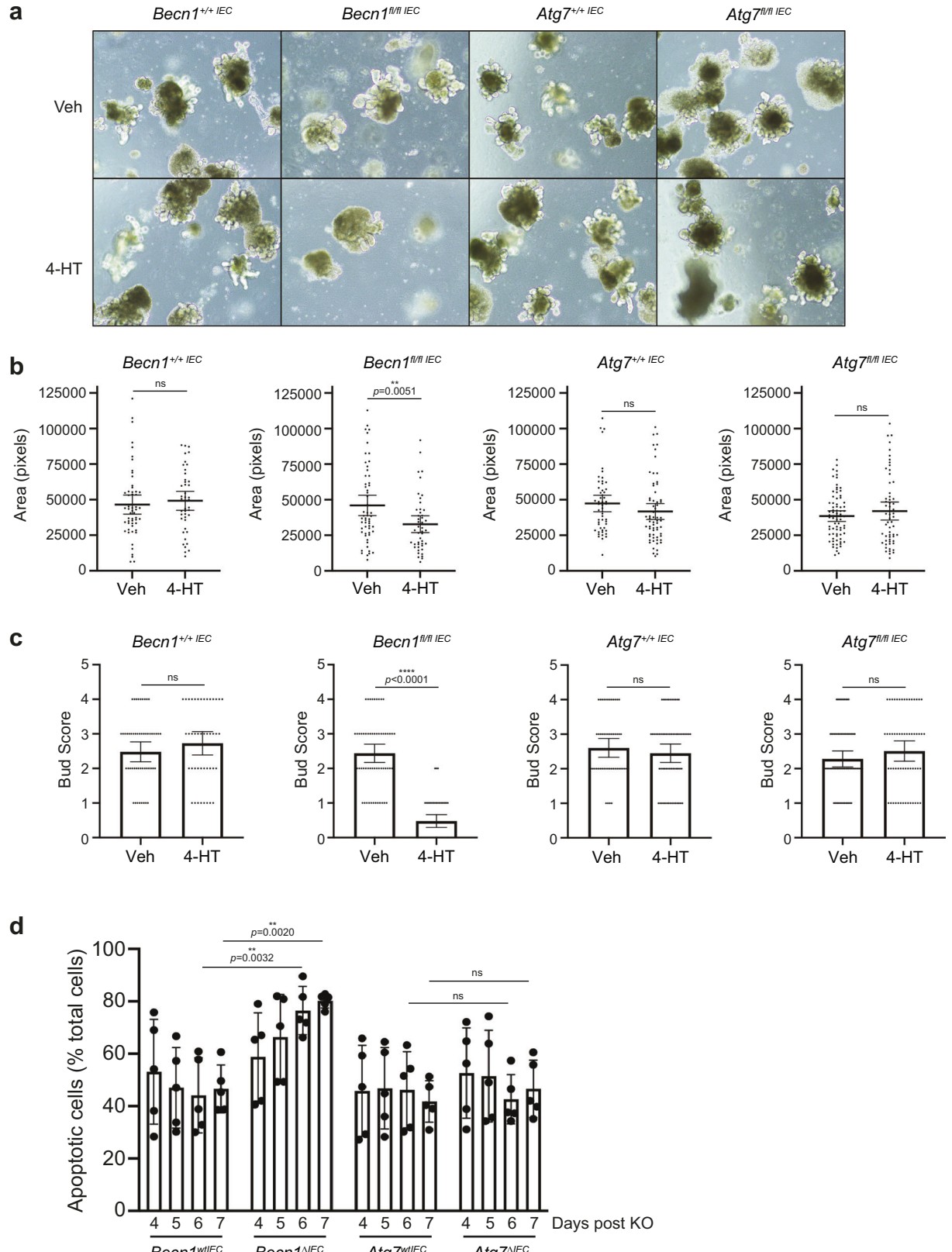

Fig. 6a, 7). However, unlike RAB5$^{+ve}$ vesicles, there was no change in the number of EEA1$^{+ve}$ vesicles (Supplementary Fig. 6b) but they were reduced in size (Supplementary Fig. 6c). Critically, there was no increased colocalisation of E-CADHERIN (Supplementary Fig. 6a, d, e, 7). This is consistent with previously published observations in neurons where the absence of

BECLIN1 leads to the impaired recruitment of EEA1 to RAB5$^{+ve}$ early endosomes[30].

We next examined late endosomes marked by RAB7 (Fig. 5f, Supplementary Fig. 8). Here we also observed a significant increase in the number of RAB7$^{+ve}$ endosomes (Fig. 5g) though unlike RAB5$^{+ve}$ or EEA1$^{+ve}$

**Fig. 4 | BECLIN1-deficient intestinal organoids display increased apoptosis and morphological defects. a, b** Deletion of BECLIN1, but not ATG7, by addition of 4-hydroxytamoxifen (4-HT) led to significantly smaller intestinal organoids. **a, c** There was also a significant reduction in the number of "buds" per organoid formed, indicative of reduced stem cell-containing crypt formation. **d** BECLIN1- and ATG7-deficient organoids were analyzed for propidium iodide (PI) and Annexin V staining by flow cytometry, to detect apoptotic cell death, over days 4 to 7 post the first 4-HT dose. Consistent with our in vivo observations, we saw enhanced apoptosis in the *Becn1*^ΔIEC^ but not *Atg7*^ΔIEC^ or wild-type organoids. Data are representative of at least *n* = 3 independent experiments. Graphs indicate the mean ± 95% confidence interval, except for the FACS plot in **d** which indicate mean ± S.D. Significance determined by Welch's unpaired *t* test for **b** and **c**, and by 2way ANOVA Tukey's multiple comparisons test for **d**. The ^+/+ IEC^ and ^fl/fl IEC^ refer to intestinal organoids derived from *Becn1*^+/+^;*Vil1-CreERT2*^Cre/+^ or *Atg7*^+/+^;*Vil1-CreERT2*^Cre/+^ and *Becn1*^fl/fl^;*Vil1-CreERT2*^Cre/+^ or *Atg7*^fl/fl^;*Vil1-CreERT2*^Cre/+^ mice.

early endosomes, there was no significant change in the size of these vesicles (Fig. 5h). This suggests potential compensatory mechanisms or delayed maturation along the endocytic trafficking pathway, as a result of the upstream defects observed with RAB5^+ve^ and EEA1^+ve^ early endosomal maturation. We also saw a significant increase in E-CADHERIN co-localization with RAB7^+ve^ late endosomes in the absence of BECLIN1, again likely consistent with the disruptions in the maturation or trafficking of late endosomes (Fig. 5f, i, j, Supplementary Fig. 8).

As described above, the correct localization of E-CADHERIN has also been reported to be regulated by RAB11^+ve^ recycling endosomes (Fig. 5k, Supplementary Fig. 9). Whilst there was no significant change in the number of RAB11^+ve^ vesicles (Fig. 5l), they appeared smaller (Fig. 5m) and lost their distinctive apical distribution to a more cytoplasmic one towards the apical end in the *Becn1*^ΔIEC^ organoids (Fig. 5k, Supplementary Fig. 9). There was also no increased co-localization of E-CADHERIN with the aberrantly distributed recycling endosomes in these *Becn1*^ΔIEC^ IECs (Fig. 5k, Supplementary Fig. 9).

Notably ATG7 deletion did not impact E-CADHERIN localization or have the same deleterious impact on the endocytic trafficking pathway in the way that BECLIN1 deletion did (Fig. 5, Supplementary Fig. 6). We also noted that neither BECLIN1 nor ATG7 loss altered total cellular E-CADHERIN levels (Supplementary Fig. 10) suggesting that it is the localization of E-CADHERIN that is compromised as a consequence of defective endocytic trafficking in the absence of BECLIN1 but not ATG7. We also saw significant changes in the average size of EEA1^+ve^, RAB7^+ve^ and RAB11^+ve^ vesicles in *Atg7*^ΔIEC^ distinct to *Becn1*^ΔIEC^ organoids (Fig. 5, Supplementary Fig. 6).

As such, the mislocalization of E-CADHERIN due to defects in the endocytic trafficking pathway, particularly in RAB5^+ve^ early endosomes, accompanied by the compromised intestinal barrier integrity, in the absence of BECLIN1 but not ATG7, highlights the contribution of BECLIN1-mediated endocytic trafficking in maintaining intestinal homeostasis.

## Discussion

We have shown that intestinal epithelial-specific BECLIN1 loss in adult mice results in their rapid and fatal deterioration due to an intestinal pathology marked by intestinal epithelial cell death, inflammation and loss of barrier integrity. For the first time, we demonstrate that BECLIN1 is critically required for the survival and specialized functions of multiple IEC subtypes. These include the maintenance of tuft cells and goblet cells, the regulation of secretory granules in Paneth cells, and the expression and localization of IAP in enterocytes. Together, these IEC defects, brought about by the loss of BECLIN1, compromise the intestinal barrier function and overall defense mechanisms of the gut. The divergence of phenotypes between *Becn1* loss, and loss of the autophagy regulator *Atg7* in the intestinal epithelium, also led us to postulate that the role of BECLIN1 in endocytic trafficking, which ATG7 is not known to be involved in[23], may at least be in part driving these critical functions in the IECs.

Notably, we show that in contrast to ATG7, BECLIN1 plays a pivotal role in maintaining an intact endocytic trafficking pathway. Consistent with previous studies[30], we observed that BECLIN1 loss primarily contributes to the disruption of RAB5^+ve^ endosome formation and the impairment of early endosome maturation. This subsequently leads to potential compensatory and adaptive mechanisms in late and recycling endosomal dynamics. We identified that a fully functional endocytic trafficking pathway mediated by

BECLIN1 is required for appropriate E-CADHERIN distribution to preserve intestinal permeability. Indeed, loss of cell-cell adhesion *via* the adherens junction has previously been shown to be critical for epithelial barrier integrity in the gut. Taken together, our data now convincingly provide the first known indicators of a physiologically essential role for BECLIN1 in a mammalian intestinal system.

The connection between endocytic trafficking and intestinal physiology has not been widely researched. Our data here characterizing *Atg7* intestinal epithelial-specific knock-out mice are consistent with published roles of autophagy in the intestinal epithelium, where a loss of autophagy conjugation proteins including ATG5, ATG16 and ATG7, results in Paneth and goblet cell defects, minimally affects enterocytes, and mice do not develop enteritis unless a chemical or pathogenic agent is administered[1,14]. Interestingly, we did not observe apoptosis and loss of intestinal stem cells in our ATG7-deficient mice, as reported in Trentesaux et al[11], but this is potentially due to our assessment of mice at a much earlier time point. However, whilst overlapping features are apparent when comparing *Atg7*- and *Becn1*-deficient mice in terms of Paneth and goblet cell changes, *Becn1*-deficient mice have dramatic differences suggestive of non-autophagic roles that are consistent with known roles of other PI3KC3 complex members.

Indeed, there is also some data from non-mammalian organisms that suggest endocytic trafficking could play an important role in the intestine. For example, knock-down of *Uvrag* in *Drosophila* ISCs was associated with impaired differentiation, intestinal dysplasia, and reduced lifespan, attributed to endocytic trafficking defects[34]. Mislocalised IEC E-cadherin and postnatal lethality with IBD-like features were observed in *Pik3c3*-deficient zebrafish[35]. In mammalian systems, intestinal epithelial loss of either *Atg14* (or *Fip200*) in mice results in a similar villi atrophy and apoptosis phenotype to *Becn1*-deficient mice, however, this took six weeks to manifest[36], probably due to the PI3KC3-C1-specific nature of ATG14, whilst BECLIN1 acts in both PI3KC3-C1 and PI3KC3-C2. Endocytic trafficking also contributes to goblet cell degranulation, requiring endolysosomal convergence with autophagosomes[5]. In enterocytes, the recycling endosome proteins RAB8 and RAB11 are necessary for the localization of apical peptidases, transporters and toll-like receptors (TLR), where loss results in intracellular vacuoles, mislocalised proteins, and for *Rab11a*, spontaneous enteritis resembling our *Becn1*-deficient mice phenotype[37,38]. However, differences in phenotype do exist, specifically deficiency in RAB8 or RAB11 is associated with the formation of microvillus inclusion bodies[37,38] which is not apparent in *Becn1*-deficient mice, suggesting different mechanistic pathways between BECLIN1 and these RAB proteins. However, we did observe evidence of defective endocytic trafficking in the phenotype of the *Becn1*-deficient mice.

The delineation between the autophagy and endocytic trafficking roles of BECLIN1, and their impacts on various IEC subtypes and barrier integrity may yield important knowledge for our understanding of basic intestinal biology and pathophysiology. Indeed, the phenotype of our BECLIN1-deficient mice is strongly reminiscent of the intestinal damage that features in inflammatory bowel disease (IBD). Although BECLIN1, unlike other autophagy regulators, has not previously been linked to IBD, our data reveal that it is a hitherto unknown essential *master regulator* of intestinal homeostasis controlling multiple facets critical for this process. Hence, further studies examining a potential role for BECLIN1 in IBD and its therapeutic modulation with compounds that lead to its increased levels or activity[39,40] are warranted.

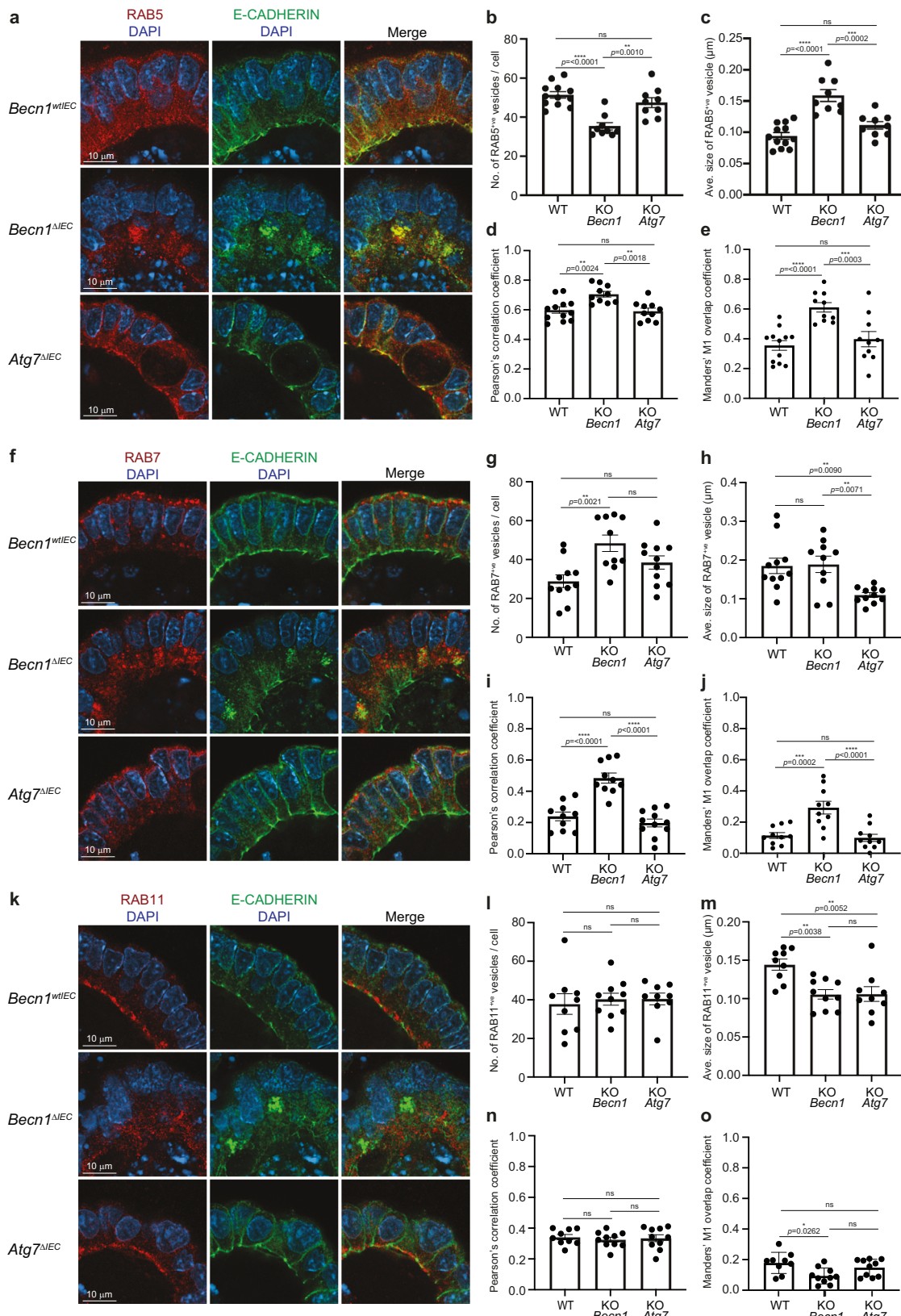

## Methods

### Mouse studies

All mouse strains used in this study are bred on the C57BL/6 J background. Becn1[tm1b(KOMP)Wtsi] mice were purchased from the European Conditional Mouse Mutagenesis Program (EUCOMM). *Becn1[fl/fl]* mice were generated by breeding Becn1[tm1b(KOMP)Wtsi] mice onto CAG-FLPe mice. These, and the *Atg7[fl/fl]* (Atg7[tm1Tchi]) mice[41] were provided by the Kile Laboratory (Garvan Institute of Medical Research, NSW, Australia). The *Vil1-CreERT2* mice[42] were provided by the Mariadason Laboratory (Olivia Newton-John Cancer Research Institute). Mice were housed at the La Trobe Animal Research and

**Fig. 5 | BECLIN1 mediates the correct localization of E-CADHERIN through its regulation of endocytic trafficking. a, f, k** The absence of BECLIN1, but not ATG7, leads to loss of lateral, basal, and apical membrane staining of E-CADHERIN as detected by whole-mount immunofluorescent staining of wild-type, *Becn1^ΔIEC^*, and *Atg7^ΔIEC^* intestinal organoids. This is consistent with the increased intestinal permeability phenotype observed in *Becn1^ΔIEC^* mice (Fig. 1k). We also characterized the endocytic trafficking pathway by immunofluorescence staining of protein markers of the various components within the pathway, namely **a–e** RAB5^+ve^ early endosomes, **f–j** RAB7^+ve^ late endosomes, and **k–o** RAB11^+ve^ recycling endosomes. In all *Becn1^ΔIEC^*-derived compartments, we observed mislocalisation of these vesicles with significant changes to the numbers and sizes of the vesicles in some compartments. Notably, the trapped E-CADHERIN within aberrantly large RAB5^+ve^ early endosomes and its mislocalisation in BECLIN1-deficient intestinal epithelial cells can be attributed to defective endocytic trafficking. Data are representative of at least n = 3 different slices per organoid and of at least n = 3 biologically independent organoids from n = 3 independent experiments. Graphs indicate the mean ± S.E.M. Significance was determined by ordinary one-way ANOVA for endosomal numbers, size, and Pearson's correlation coefficient and by two-way ANOVA for the Mander's M1 overlap coefficient.

Teaching Facility (LARTF, La Trobe University, VIC, Australia) under Specific Pathogen Free (SPF) conditions. To induce deletion, male and female mice used indiscriminately, aged six weeks or older were intraperitoneally injected with 4 mg tamoxifen (Sigma-Aldrich, T5648) in sunflower seed oil (Sigma-Aldrich, 25007), delivered as one injection per day of 200 μL of a 10 mg/mL stock, over two consecutive days. Mice were humanely endpointed by $CO_2$ asphyxiation. For PCR genotyping, for colony maintenance and KO assessment, DNA extracts were prepared from ear clips or isolated IECs (see IEC isolation) by their overnight incubation at 56 °C followed by 1 hour at 85 °C in DirectPCR (Tail) Lysis Reagent (Viagen) supplemented with 0.2% (v/v) Proteinase K (Sigma-Aldrich). PCR reactions were prepared with GoTaq PCR Master Mix (Promega) following the manufacturer's protocol. Primers for *Becn1* reactions are as follows: "floxed forward"—5′ CTG ATC CTG CAG CTT GCA GAT TAG C3′, "floxed reverse"—5′CAC CAC TGC CTG GCT AAA CAA GAG C3′, "KO reverse"—5′CTA TAG AAG AAA AGG ACT GTT GTG AC3′. Primers for *Atg7* reactions are as follows: "floxed forward"—5′TGG CTG CTA CTT CTG CAA TGA TGT3′, "floxed reverse"—5′CAG GAC AGA GAC CAT CAG CTC CAC3′, "WT forward"—5′TCT CCC AAG ACA AGA CAG GGT GAA3′, "WT reverse"—5′AAG CCA AAG GAA ACC AAG GGA GTG3′. Primers for *Vil1-CreERT2* reactions are as follows: forward—5′CAA GCC TGG CTC GAC GGC C3′, reverse - 5′CGC GAA CAT CTT CAG GTT CT3′. For intestinal barrier permeability experiments, mice were fasted overnight and then orally gavaged with 150 μL of 10 mg/mL FITC-dextran (Sigma-Aldrich, FD4) in DPBS (Gibco). Blood was collected into Microvette® EDTA K tubes (Sarsdtedt) both prior to gavage *via* submandibular bleed as a baseline, and 4 hours after gavage *via* terminal cardiac puncture. Plasma was obtained from the supernatant fraction after centrifugation of blood at 2300 × g. The fluorescence of plasma was measured by the EnSight™ Plate Reader (PerkinElmer). All experiments performed were approved by the La Trobe University animal ethics committees (AECs, AEC18024, AEC18036) in accordance with the Australian code for the care and use of animals for scientific purposes. We have complied with all relevant ethical regulations for animal use.

## IEC isolation
Intestinal sections were dissected open, rinsed in DPBS, and incubated in 15 mM EDTA in DPBS for 15 mis at 37 °C with agitation. IECs were dissociated into a solution by vortexing. Cells were washed once in chilled DPBS and snap-frozen over dry ice, stored at −80 °C until use.

## Intestinal organoid culture
Organoids were established by culturing crypt-enriched fractions from the duodenum of untreated mice. To isolate crypt-enriched fractions, duodenal tracts were opened and rinsed in DPBS. Villi were removed by gentle scraping. Tracts were cut into 0.5 cm sections, rinsed vigorously with DPBS, and incubated in 2 mM EDTA in DPBS for 30 minutes at 4 °C with gentle agitation. Sections were washed once in cold DPBS and crypts obtained by dissociation from the stromal tissue into DPBS by vigorous pipetting. Crypts were resuspended in Advanced DMEM media (Advanced DMEM (Gibco), 1% (w/v) bovine serum albumin, 20 mM L-Glutamine, 100 mM HEPES, 1000 U/mL Penicillin-Streptomycin) and passed through a 70 μm strainer. Remaining single cells were excluded by repeat low speed column centrifugation (76 × g, 2 minutes) and discarding of cloudy supernatant. Crypt-

enriched pellets were then resuspended in Cultrex Reduced Growth Factor Basement Membrane Extract, Type 2, Path Clear (R&D Systems) at 50-300 crypts per 50 μL dome in pre-warmed 24 well plates. Domes were polymerized by incubation for 15 minutes at 37 °C and Advanced DMEM media supplemented with 1× N-2 (Gibco), 1× B-27 (Gibco), 0.05 ng/μL human EGF (Peprotech, AF-100-15), 0.1 ng/μL murine noggin (Peprotech, 250-38), 0.5 ng/μL murine R-Spondin-1 (Peprotech, 315-32) was added. Organoids were maintained at 37 °C, 5% $CO_2$, with media being replaced every 2–3 days. Passaging was performed every 7–10 days by mechanically dissociating organoids and re-seeding into fresh Cultrex at a 1:3 to 1:6 split. To induce deletion, organoids were seeded into media containing 200 nM 4-HT (Sigma-Aldrich, H7904) for three days, and then maintained as per normal.

## Western immunoblotting
Cells were lysed on ice for 1 hour in 20 mM Tris pH 7.5, 135 mM NaCl, 1.5 mM $MgCl_2$, 1 mM EDTA, 10% (v/v) glycerol, 1% (v/v) Triton X-100, supplemented with cOmplete Protease Inhibitor Cocktail (Roche, following the manufacturer's protocol). Lysates were centrifuged for 5 minutes at 16,000 × g and supernatant protein concentrations determined using the Pierce BCA Protein Assay Kit (Thermo Scientific) according to the manufacturer's protocol. Equalized amounts of protein between 15 and 40 ng per lane were boiled for 5 minutes at 95 °C in reducing buffer (4× stock prepared at 178.3 mM Tris, 350 mM dithiothreitol, 288.4 mM SDS, 670 mM bromophenol blue, 36% (v/v) glycerol, 10% (v/v) β-mercaptoethanol) and electrophoresed on 4–12% polyacrylamide NuPAGE™, Bis-Tris Mini Protein Gels (Invitrogen) using NuPAGE™ MES SDS Running Buffer (Invitrogen). Proteins were wet transferred in NuPAGE™ Transfer Buffer supplemented with 10% (v/v) methanol, using the Mini Blot Module (Invitrogen) at 17 V for 1 hour, onto 0.45 μm nitrocellulose membranes (Amersham Protran). Membranes were blocked in 5% (w/v) skim milk in PBS (2.7 mM KCl, 1.76 mM $KH_2PO_4$, 136.7 mM NaCl, 8.07 mM anhydrous $Na_2HPO_4$) for 1 hour at room temperature (RT) with agitation. Membranes were probed overnight at 4 °C with the following antibodies at the following ratios, prepared in 1% (w/v) skim milk in PBST (PBS + 0.05% (v/v) Tween-20): Beclin1, 1:500 (CST, 3495); ATG7, 1:500 (Sigma-Aldrich, A2856), p62/SQSTM1, 1:500 (CST, 5114); LC3B, 1:500 (Novus Biologicals, NB100-2220); β-Actin, 1:5000 (Sigma-Aldrich, A2228); E-cadherin, 1:600 (CST, 3195); GAPDH, 1:5000 (Invitrogen, MA5-15738). Membranes were washed in PBST and probed for 1 hour at RT with the following antibodies, prepared in 1% (w/v) skim milk in PBST: Donkey anti-Rabbit IgG, 1:10,000 (GE Healthcare, NA943V), Goat anti-Mouse IgG, 1:10,000 (Sigma-Aldrich, A0168). Luminescent signals were visualized using the Western Lightning Plus-ECL (PerkinElmer) kit and ChemiDoc Imaging System (Bio-Rad). Images were processed using Image Lab (Bio-Rad).

For the detection of E-CADHERIN in intestinal organoids, the organoids were first dissociated from Cultrex by resuspending them in ice-cold DPBS, centrifuged at 300 × g for 2 min, and manually aspirating as much of the Cultrex layer as possible. This step was repeated as necessary until most of the Cultrex was removed. Lysis buffer (50 mM Tris-HCl pH 6.8; 2% SDS; 1.5 mM Bromophenol blue; 10% glycerol; 100 mM DTT) supplemented with cOmplete Protease Inhibitor Cocktail (Roche, following the manufacturer's protocol) was then added, and the organoids were homogenized by vigorous pipetting. The lysates were then heated at 95 °C for 10 minutes,

and protein concentration determined by measuring the absorbance at 280 nm using a NanoDrop Spectrophotometer. Equal amounts of proteins were loaded into each lane prior to gel electrophoresis, and the subsequent downstream steps were performed as outlined above.

## Immunohistochemistry and histology

Formalin-fixed, paraffin-embedded intestinal sections of 4 μm thickness were dewaxed with xylene and rehydrated into RO water *via* decreasing ethanol grades. For immunohistochemistry, antigen retrieval was performed by boiling slides in citrate buffer (10 mM sodium citrate, 0.05% (v/v) Tween-20) for 20 minutes. Slides were quenched for 20 minutes in 3% (w/w) hydrogen peroxide (ChemSupply, HA154-500M) and rinsed in TBST (0.5 M Tris, 9% (w/v) NaCl, 0.5% (v/v) Tween-20, pH 7.6). Sections were incubated overnight in a humidified chamber with primary antibodies diluted in SignalStain® Antibody Diluent (Cell Signalling Technology (CST), 8112) at the following ratios: Cleaved caspase-3, 1:200 (CST, 9661); Ki-67, 1:150 (Invitrogen, MA5-14520); Lysozyme, 1:300 (Thermo Scientific, RB-372-A1); Chromogranin A, 1:500 (Abcam, ab85554); DCAMKL1, 1:400 (Abcam, ab31704). Slides were washed in TBST, and antigens were detected using the EnVision+ System (DAKO, K4003) following the manufacturer's protocol. Slides were counterstained with Mayer's hematoxylin, dehydrated in increasing ethanol grades, cleared in xylene, and mounted with DPX (Sigma-Aldrich). For TUNEL staining, the TUNEL Assay Kit (Abcam, ab206386) was used according to the manufacturer's protocol. For IAP staining, slides were dewaxed and rehydrated into PBST (see Western immunoblotting), and the Vector Red Alkaline Phosphatase Substrate Kit (Vector Laboratories SK-5100) was used according to the manufacturer's protocol and included counterstaining with Mayer's haematoxylin. For PAS-AB staining, dewaxed and rehydrated slides were incubated for 15 minutes in 1% (v/v) Alcian blue in 3% (v/v) aqueous acetic acid (Amber Scientific), 5 minutes in 1% (v/v) aqueous period acid (Amber Scientific), and 10 minutes in Schiff's reagent (Amber Scientific, with RO water rinses in between steps. Slides were also counterstained with Mayer's haematoxylin.

## Transmission electron microscopy

Duodenal tissue was fixed in Karnovsky's fixative (2.5% (v/v) glutaraldehyde, 2% (v/v) paraformaldehyde, 0.1 M sodium cacodylate) for 4 hours at 4 °C. Tissue was post-fixed in 1% (w/v) osmium tetroxide, 1.5% (w/v) potassium ferrocyanide and 0.1 M sodium cacodylate, for 3 hours at RT. Tissues were rinsed in Milli-Q water and dehydrated in increasing grades of ethanol. Samples were embedded into Spurrs resin by two immersions in 100% acetone for 20 minutes each, then 2 hour immersions in mixtures of Spurrs to acetone at 1:3, 1:1, and 3:1 ratios. Overnight infiltration in 100% Spurrs was followed by 2 hours in 100% Spurrs under vacuum conditions. Blocks were polymerized overnight at 70 °C. 70 nm ultrathin sections were contrasted for 10 minutes in 10% (w/v) uranyl acetate in methanol under dark conditions, rinsed sequentially with 50% (v/v) methanol, 25% (v/v) methanol then Milli-Q water, followed by incubation in Reynold's lead citrate solution (0.11 M lead nitrate, 0.19 M sodium citrate, pH 12, prepared in $CO_2$-free Milli-Q water) for 10 minutes in carbon dioxide depleted conditions. Sections were mounted onto a copper grid, carbon-coated, and imaged with the Joel JEM-2100 transmission electron microscope at 80 kV equipped with Gatan digital camera.

## Organoid cell death assays

For apoptosis assays, organoids were dissociated by incubation for 15 minutes at 37 °C in TrypLE Express (Gibco) plus pipetting. Cells were stained in the dark for 15 minutes with either FITC Annexin V (BD Pharmingen, 556419) or APC Annexin V (BD Pharmingen, 550475) diluted at 1:20, and Propidium iodide solution (Sigma-Aldrich, P4864) diluted at 1:200 in 1× Annexin V Binding Buffer (BD Pharmingen, 556454). Stained cells were diluted 4× in 1× Annexin V Binding Buffer and analyzed by flow cytometry performed using the BD FACSCanto II (Becton Dickinson and Company, BD Biosciences, San Jose, CA, USA). Data was analyzed using FlowJo™ Software, Version 10.5.3 (Becton Dickinson and Company, BD Biosciences, San Jose, CA, USA).

## Intestinal organoid whole-mount immunofluorescence

Whole-mount intestinal organoid staining protocol was adapted from Dekkers et al.[43,44]. Briefly, organoids were removed from matrix by incubation with ice-cold Gentle Cell Dissociation Reagent (STEMCELL Technologies, 100-0485) with gentle rocking at 4 °C for 60 minutes. Organoids were then fixed in 4% (w/v) paraformaldehyde solution (ProSciTech, C004) and blocked with Organoid Wash Buffer (DPBS (Gibco™, 14190144) + 0.1% (w/v) Triton X-100 (Merck, X-100) + 0.2% (w/v) Bovine Serum Albumin (Merck, A3059)), followed by overnight incubation at 4 °C with gentle rocking using the following primary antibodies at the following dilutions: E-CADHERIN, 1:500 (Invitrogen, 13-1900); RAB5, 1:100 (CST, 46449); EEA1, 1:100 (CST, 3288); RAB7, 1:100 (CST, 9367); RAB11, 1:100 (CST, 5589). Following primary antibody incubation, organoids were washed extensively and incubated overnight at 4 °C with gentle rocking using the following secondary antibodies: Goat anti-Rat IgG (H + L) cross-adsorbed secondary antibody, Alexa Fluor™ 568 (1:500, Invitrogen, A-11077); Goat anti-Mouse IgG (H + L) cross-adsorbed secondary antibody, Alexa Fluor™ 488 (1:500, Invitrogen, A-11004); Goat anti-Rabbit IgG (H + L) Cross-Adsorbed Secondary Antibody, Alexa Fluor™ 488 (1:500, Invitrogen, A-11008) and nuclear stain DAPI (1 μg/ml, Merck, D9542). Organoids were then subjected to another extensive washing step prior to sample clearing and mounting steps. In lieu of the fructose-glycerol solution used for organoid clearing and mounting as outlined in Dekkers et al.[43], RapiClear® 1.49 solution (SunJin Lab, RC149001) was used instead in accordance with the manufacturer's protocol. Images were then acquired using Zeiss LSM 980 with Airyscan 2 confocal microscope. Airyscan 2 Multiplex SR-4Y imaging mode was utilized, and parameters such as laser power and gain were kept consistent amongst samples. Images were taken using the 40× (water) objective with 1.7× Zoom and imaged once to avoid excessive photobleaching.

## Quantitative image analysis

Quantitative analysis of fluorescent images was performed using ImageJ Software (Fiji). Spatial calibration was first performed (Analyze>Set Scale). For each analysis, a region of interest (ROI) comprising >5 nuclei (DAPI) was manually drawn to encompass all relevant structures while excluding non-specific luminal debris staining in the organoid lumen.

To derive the number of endosomal vesicles per cell (Fig. 5b, g, l, Supplementary Fig. 6b) and the average vesicle size (Fig. 5c, h, m, Supplementary Fig. 6c), the RAB5/EEA1/RAB7/RAB11 channel was thresholded to create a binary image containing the structures of interest (Image>Adjust>Threshold (default)) followed by the application of the watershed separation plugin (Process>Binary>Watershed) to separate the structures (referred to herein as particles). The particles within the ROI were then analyzed (Analyze>Analyze Particle) to obtain the average size of particle, and the number of particles per cell was further calculated by dividing the total count of particles by the number of nuclei.

The analysis of RAB5/EEA1/RAB7/RAB11 and E-CADHERIN colocalisation was carried out using the BioImaging And Optics Platform (BIOP) Just Another Co-localization Plugin (JACoP) (Plugin>BIOP>Image Analysis>BIOP JACoP). Manual thresholding was applied to select structures of interest, and the analysis was specifically performed on the designated ROI as described earlier. Thresholded Pearson's correlation coefficient and Mander's overlap coefficient values were then extracted from the data generated.

## Intraepithelial lymphocytes isolation for flow cytometry analysis

Intraepithelial lymphocytes were isolated from the small intestines and colons of mice as described above. Briefly, excess fat was removed from the colon and small intestines. Peyer's patches were also removed from the small intestines. The tissues were kept moist by frequent saturation with ice-cold wash buffer (dPBS + 2% (w/v) fetal calf serum). Intestines were cut

longitudinally through the lumen, and fecal matter washed away by lightly swirling in a petri dish containing ice-cold wash buffer. Intestines were then cut into 0.5 cm fragments and placed in dissociation solution (wash buffer + 5 mM EDTA), and incubated for 30 minutes at 37 °C with gentle shaking. The cell suspension was then vortexed and filtered using a 70 μm cell strainer, transferred to a new 50 mL tube and centrifuged at 1700 rpm for 7 minutes. The cell pellets were then washed twice with wash buffer. Cells were then resuspended in discontinuous 40%/80% Percoll® (GE Healthcare, 17089101) gradient and centrifuged at $900 \times g$ for 20 minutes at room temperature. The interface was collected and washed twice with wash buffer, and the final pellet containing intraepithelial lymphocytes was stained.

For flow cytometry analysis, isolated IELs from above were incubated with Live/Dead marker (1:500, BD Biosciences, 566332), anti-CD45.2 (1:200, BD Biosciences, 741957), CD3ε (1:200, BD Biosciences, 564378), CD4 (1:1000, BD Biosciences, 741461), CD8α (1:1000, BD Biosciences, 562283), CD8β (1:1000, BD Biosciences, 741127), TCRβ (1:300, Thermo-Fisher Scientific, 47-5961-82), TCRγδ (1:300, BD Biosciences, 563994), NK1.1 (1:1000, BD Biosciences, 566502), FOXP3 (1:300, Thermofisher Scientific, 12-5773-82), and CD19 (1:300, BD Biosciences, 563333) for total IEL analysis. For intracellular staining, cells were fixed and permeabilised using the Foxp3/Transcription factor staining buffer kit (Invitrogen, 00-5523-00) according to the manufacturer's instructions, followed by incubation with intracellular markers. Finally, cells were resuspended in FACS wash buffer containing CountBright Absolute Counting Beads (Molecular Probes, C36950), and stained cell acquisition was performed on the BD FACSymphony™ A3 Cell Analyzer (Becton Dickinson and Company, BD Biosciences, San Jose, CA, USA) and data analyzed using FlowJo™ Software, Version 10.5.3 (Becton Dickinson and Company, BD Biosciences, San Jose, CA, USA).

## Statistics and reproducibility
Numerical source data for all graphs are provided in Methods and Figure legends. Statistical tests were performed using GraphPad Prism 8 Software (GraphPad, San Diego, CA) via Welch's unpaired $t$ tests between two groups and one-way or two-way ANOVA for multiple comparisons. Each mouse was assessed as an individual sample. All data were obtained by performing at least $n = 3$ independent experiments with representative data shown and expressed as the mean ± standard error of the mean (S.E.M) or ± standard deviation (S.D.). $P$ values < 0.05 were considered statistically significant. Significance levels were split further as to **$P < 0.01$, ***$P < 0.001$, ****$P < 0.0001$.

## Reporting summary
Further information on research design is available in the Nature Portfolio Reporting Summary linked to this article.

## Data availability
Numerical source data are provided in the Excel file Supplementary Data 1. Source images for representative western blots are shown in Fig. 1b, Supplementary Figs. 2b, 4c, and 10 are provided in Supplementary Figs. 11–14, respectively, in Supplementary information. Gating strategies for FACS data presented in Fig. 4d and Supplementary Fig. 3 are provided in Supplementary Figs. 15 and 16. Any other data that support the findings of this study are available within the article, its Supplementary information and from the corresponding author upon reasonable request.

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

## Acknowledgements
We thank Professor Joan Heath and Dr. Karen Doggett for providing the B6.Cg-*Ndor1*^Tg(UBC-cre/ERT2)1Ejb^/1 J or UBC-Cre-ERT2 mice (Walter and Eliza Hall Institute, Melbourne, VIC, Australia). We acknowledge scholarship support for S.T. (La Trobe University Research Training Program Scholarship), L.J.J. (La Trobe University Australian Postgraduate Award), and J.J. (La Trobe Graduate Research Scholarship and Full Fee Research Scholarship). We are grateful to the Australian Research Council for grant support (E.F.L., W.D.F., J.M.M., DP190102612; P.A.G., DP160102394) and to the National Health and Medical Research Council (J.M.M., 1046092; K.D., A.S.Y., 2010704, 1136592), the Victorian Cancer Agency (E.F.L., MCRF19045) for fellowship support. J.C.L. is a Lister Institute Prize Fellow and supported by the Francis Crick Institute, which receives its core funding from Cancer Research UK (CC2219), the UK Medical Research Council (CC2219), and the Wellcome Trust (CC2219).

## Author contributions
S.T., J.J. designed, performed, and analyzed experiments and wrote the paper; T.J.H., M.E., J.R. S.L.E., C.M.R. R.N., I.Y.L., L.J.J., S.G., M.Y., C.I., D.B. designed, performed experiments, analyzed and interpreted data; C.J., M.B., J.C.L., P.D.C., K.D., P.A.G., B.T.K., L.A.M., A.S.Y., J.M.M. analyzed and interpreted data; W.D.F., E.F.L. designed the project, analyzed, interpreted data, and wrote the paper. All authors commented on the manuscript.

## Competing interests
The authors declare no competing interests.
