## [Peer review file · Communications Biology]

Reviewers' comments:

Reviewer #1 (Remarks to the Author):

The manuscript entitled "BECLIN1 is essential for intestinal homeostasis" reports the analysis of the effects of the intestinal epithelium-specific BECLIN1 deletion in adult mice. The phenotype obtained is compared with that caused by the ATG7 deletion, showing distinct functions of the two autophagy regulators and a new Beclin1 function for the maintenance of intestinal homeostasis. In my opinion, the manuscript is convincing and well written. The conclusions are original. Methods are described in sufficient details. I have just some suggestions.

59: variants in, autophagy regulators: variants in autophagy regulators.

65: Whilst these are important roles for autophagy regulators in the maintenance of intestinal homeostasis, autophagy-deficient intestines generally function normally, and autophagy deficiency alone is insufficient for driving spontaneous intestinal pathologies: please, add reference(s).

Figure 2 and Figure 3.a: Pay attention to the units of measurement of length. The symbol for micrometer is μm , not μM .

Figure 2.b: there are three figures showing the same type of specimen (Becn1 Δ IEC) with evidence of abnormal multivesicular body accumulation. Please, add a control showing normal multivesicular bodies (Becn1wtIEC or Atg7 Δ IEC).

177: In contrast, there were no significant changes to the numbers of enteroendocrine (ChgA+ve, Fig.2e) or Paneth subpopulations: Paneth subpopulations are not shown in Figure.

325: Please specify if all mice were generated by breeding C57BL/6J strains.

342: "KO reverse": please, check the primer and primer name.

515: We thank Prof. Joan Heath and Dr. Karen Doggett for providing the UBC-Cre-ERT2 mice: check the mice strain.

Reviewer #2 (Remarks to the Author):

The manuscript by Tran and collaborators describes and demonstrates, very well, that BECLIN1 is essential for gut homeostasis bringing data both in vivo and on organoids. The research is well thought out, the manuscript is well written. It achieves interesting and new results. I am only giving some suggestions for improving the manuscript and some indications of typos to correct.

Suggestions for improving the manuscript:

Today it is quite well known that the expression of a gene can be missing both because the gene is deleted but also because the gene is epigenetically silenced, for example, by methylation of its promoter. A very recent article, which reports the epigenetic modulation power of the phytochemical indicaxanthin (which induces autophagy) on intestinal cells, describes the methylomic effects on the BECN1 gene and on other genes of this molecule extracted from the prickly pear *Opuntia Ficus Indica* (Ragusa et al., 2023; PMID: 37571432). I propose that in the discussion, in the final part, the authors can show the comparison with what indicaxanthin epigenetically does on intestinal cells in vitro, also regarding autophagy, and therefore take the opportunity to relaunch new future studies in the direction of epigenetics.

Minor fixes:

All gene acronyms have to be reported, according to international standardizations, in capitals and italics. In lines 271 and 277 the abbreviations of the BECN and ATG genes are in lower case.

Reviewer #3 (Remarks to the Author):

The authors investigate the role of BECLIN1 in gut maintenance. They demonstrate that intestinal epithelium-specific BECLIN1 deletion, but not ATG7, lead to rapid fatal enteritis with compromised gut barrier integrity. The authors then highlight the contribution of BECLIN1-mediated endocytic trafficking in the mislocalisation of E-cadherin. Overall, conclusions reached are consistent with the results obtained. However, I had some concerns that were raised through the reading of this manuscript.

Major concerns:

The authors should show additional lines of evidence to suggest that the mislocalisation of E-cadherin is primarily due to BECLIN1-mediated endocytic trafficking.

Minor concerns:

1. In Figure 1b and Supplementary Figure 2b, loss of ATG7 did not result in an obvious accumulation of p62, unlike loss of BECLIN1.

2. Figure 1i and k look blurry.

3. Labeling of "ns" is not uniform.

We thank the reviewers for their comments which we have now addressed (as listed below), and which we believe have improved our article. All changes made can be seen in the tracked change version of the resubmitted manuscript, which was provided alongside the “clean” resubmitted version.

REVIEWER #1 (REMARKS TO THE AUTHOR):

59: variants in, autophagy regulators: variants in autophagy regulators.

This has now been changed.

65: Whilst these are important roles for autophagy regulators in the maintenance of intestinal homeostasis, autophagy-deficient intestines generally function normally, and autophagy deficiency alone is insufficient for driving spontaneous intestinal pathologies: please, add reference(s).

Reference to Foerster EG *et al* (2022), *Autophagy* 18:86-103 has now been added (ref. 14).

Figure 2 and Figure 3.a: Pay attention to the units of measurement of length. The symbol for micrometer is μm , not μM .

These have all now been changed to μm .

Figure 2.b: there are three figures showing the same type of specimen (Becn1 Δ IEC) with evidence of abnormal multivesicular body accumulation. Please, add a control showing normal multivesicular bodies (Becn1wtIEC or Atg7 Δ IEC).

We were unable to find suitable images of publication quality with definitive multivesicular bodies in the transit-amplifying compartment of either Becn1wtIECs or Atg7 Δ IECs in our existing data set. Unfortunately, due to the team member responsible for these images leaving our group, and limited funding to carry on this aspect of the work, we no longer have the capacity to obtain more TEM images to address this comment. The original intention of including this figure was to provide some evidence of defective endocytic trafficking in Becn1 Δ IEC IECs (beyond the (mis)localisation data for E-CADHERIN). However, with the inclusion of the extensive new data (as requested) in Figure 5 and Supplementary Figures 5-9 (see Reviewer 3 response) where we have detailed the changes in various endosomal compartments, we believe we have now more convincingly addressed the original point intended with the TEM images in the original Figure 2b. As such, we have removed Figure 2b and updated associated figure legends.

177: In contrast, there were no significant changes to the numbers of enteroendocrine (ChgA+ve, Fig.2e) or Paneth subpopulations: Paneth subpopulations are not shown in Figure.

We have now included Figure 2e, and updated the figure legend, to provide data pertaining to the Paneth cell subpopulation, indicated by positive lysozyme staining.

325: Please specify if all mice were generated by breeding C57BL/6J strains.

We have now amended to include line: “All mouse strains used in this study are bred on the C57BL/6J background.”

342: “KO reverse”: please, check the primer and primer name.

This has been checked and is accurate. No changes made.

515: We thank Prof. Joan Heath and Dr. Karen Doggett for providing the UBC-Cre-ERT2 mice: check the mice strain.

This has been checked and is correct. However, as UBC-CreERT2 is the common name, we have now also included the full nomenclature of this mouse strain which is B6.Cg-*Ndor1*^{Tg(UBC-cre/ERT2)1Ejb}/1J in the Acknowledgements.

REVIEWER #2 (REMARKS TO THE AUTHOR):

Suggestions for improving the manuscript:

I propose that in the discussion, in the final part, the authors can show the comparison with what indicaxanthin epigenetically does on intestinal cells in vitro, also regarding autophagy, and therefore take the opportunity to relaunch new future studies in the direction of epigenetics.

We have taken on Reviewer 2's suggestion to improve the manuscript by including the following text in the closing paragraph of the discussion, with reference to papers describing agents that can increase the activity or levels of Beclin1 - Ragusa *et al.* (2023) and Shoji-Kawata *et al.* (2013):

“Hence, further studies examining a potential role for BECLIN1 in IBD and its therapeutic modulation with compounds that lead to its increased levels or activity^{39,40} are warranted.”

Minor fixes:

All gene acronyms have to be reported, according to international standardizations, in capitals and italics. In lines 271 and 277 the abbreviations of the BECN and ATG genes are in lower case.

As the gene references implied refer to mouse genes, according to the HUGO Gene Nomenclature Committee, these should begin with an upper-case letter and followed by lower-case letters. As such, no changes were made to gene names reported. However, we have altered E-cadherin to E-CADHERIN where reference is made to the mouse protein.

REVIEWER #3 (REMARKS TO THE AUTHOR):

Major concerns:

The authors should show additional lines of evidence to suggest that the mislocalisation of E-cadherin is primarily due to BECLIN1-mediated endocytic trafficking.

We have now included an extensive set of new experiments examining the various compartments within the endocytic trafficking pathway and the impact of BECLIN1 or ATG7 loss using wild-type, BECLIN1-deficient and ATG7-deficient intestinal organoids. In addition, we also examined E-CADHERIN localisation and abundance in the context of these intestinal epithelial cells. These new results are presented in Figure 5 and Supplementary Figures 5 to 10. We have also

significantly expanded upon the section “BECLIN1 loss leads to mislocalisation of E-CADHERIN due to defects in the endocytic trafficking pathway” to include this new data. Updates have also been made where needed to the Materials and Methods and Figure Legends.

Although we had originally planned to prepare a separate manuscript on this point, we thank the reviewer for this suggestion as we believe it has significantly strengthened the conclusions of this current manuscript demonstrating an important role of BECLIN1-mediated endocytic trafficking in the regulation of intestinal homeostasis.

Minor concerns:

1. In Figure 1b and Supplementary Figure 2b, loss of ATG7 did not result in an obvious accumulation of p62, unlike loss of BECLIN1.

We agree that as compared to loss of BECLIN1, loss of ATG7, does not lead to the same obvious degree of p62 accumulation in IECs. However, the complete lack of LC3-I to LC3-II conversion in the ATG7-deleted IECs provides strong evidence of defective autophagy signalling as expected.

2. Figure 1i and k look blurry.

We have now fixed the resolution such that these are sharper.

3. Labeling of "ns" is not uniform.

We have now indicated “ns” in all figures where appropriate.

REVIEWERS' COMMENTS:

Reviewer #1 (Remarks to the Author):

Second revision "BECLIN1 is essential for intestinal homeostasis"

Authors carefully revised their manuscript based on reviewers' comments and the manuscript is now significantly improved. In my opinion, the paper is now suitable for publication.

Reviewer #3 (Remarks to the Author):

The authors have addressed my concerns.